# Non-Invertible Public Key Certificates

**DOI:** 10.3390/e23020226

**Published:** 2021-02-12

**Authors:** Luis Adrián Lizama-Perez, J. Mauricio López R.

**Affiliations:** 1Dirección de Investigación, Innovación y Posgrado, Universidad Politécnica de Pachuca, Ex-Hacienda de Santa Bárbara, Zempoala, Hidalgo 43380, Mexico; 2Cinvestav Querétaro, Libramiento Norponiente 2000, Real de Juriquilla, Santiago de Querétaro, Querétaro 76230, Mexico; jm.lopez@cinvestav.mx

**Keywords:** non-invertible, cryptography, certificate, PKI

## Abstract

Post-quantum public cryptosystems introduced so far do not define a scalable public key infrastructure for the quantum era. We demonstrate here a public certification system based on Lizama’s non-invertible key exchange protocol which can be used to implement a secure, scalable, interoperable and efficient public key infrastructure (PKI). We show functionality of certificates across different certification domains. Finally, we discuss a method that enables non-invertible certificates to exhibit perfect forward secrecy (PFS).

## 1. Introduction

Since its origin in the late seventies, public key cryptography (PKC) has been exploited to support user authentication and digital signatures over the internet. In PKC, each user has two keys, the public $P_{u}$ and the private key $P_{r}$, which are mutually inverse in some mathematical sense. Not taking into account formal details we would write that $P_{r}={P_{u}}^{-1}$ thus, to achieve confidentiality, a message *m* is encrypted using Bob’s public key; symbolically we write ${\left[m\right]}_{P_{u}}$, then it is decrypted with the private key so $m={\left[m\right]}_{P_{u}{P_{u}}^{-1}}$. In contrast, to guarantee message authentication, *m* is encrypted with Alice’s private key and decrypted with her public key. Symbolically we can write it as $m={\left[m\right]}_{P_{r}{P_{r}}^{-1}}$.

Unfortunately, Shor’s algorithm [1] solves over a hypothetical quantum computer, the mathematical problems on which PKC is supported: integer factorization and discrete logarithm. In fact, most of the public key cryptosystems used today will become obsolete in the foreseeable future because they would be broken by quantum computers [2]. For this reason, the National Institute of Standards and Technology (NIST) initiated in 2015 a process to evaluate cryptographic algorithms to choose the appropriate methods for the quantum era. To this date, the selection process is in the third evaluation round [3,4].

The present work enhances a newly claimed post-quantum method called non-invertible key exchange method (ni-KEP) which was conceived to establish a secret key between two remote parties. Lizama’s ni-KEP is mathematically supported by Euler’s theorem as RSA, it uses exponentiation to exchange a secret key as Diffie–Hellman and it encrypts/decrypts through invertible multiplication as ElGamal cipher. Lizama’s non-invertible key exchange protocol was introduced in [5]. Initially, the protocol was conceived to transfer a secret value from Alice to Bob. We describe briefly the three development stages of the algorithm:1.*Multiplication-based protocol.* In a ring with unity over $Z_{n}$ where n=p·q and *p*, *q* are prime numbers. An integer may or may not have a multiplicative inverse. Multiplication between invertible and a non-invertible integer yields a non-invertible integer according to the basic properties of modular arithmetic. Alice multiplies a random non-invertible $v_{a}$ by a random invertible $k_{a}$, then she sends the result to Bob who multiplies it by his random invertible $k_{b}$ returning the resulting integer to Alice who removes $k_{a}$ multiplying by $k_{a}^{-1}$ and sending the result to Bob. Finally, Bob removes his invertible integer applying $k_{b}^{-1}$. At this point Bob has obtained $v_{a}$. Although a non-invertible integer does not have a multiplicative inverse, hence factorization of the public integers are prohibited, a division attack is discussed in [5].2.*Exponent-based protocol.* The integer that results after exponentiation say $p^{x_{a}}$ gives a non-invertible integer. Using this math property, the protocol defines that Alice sends $p^{x_{a}}k_{a}\;\mathrm{mod}\;n$ to Bob who returns $p^{x_{a}}k_{a}k_{b}\;\mathrm{mod}\;n$ to her. Then she multiplies it by $k_{a}^{-1}$ and sends back $p^{x_{a}}k_{b}\;\mathrm{mod}\;n$. Bob applies $k_{b}^{-1}$ thus obtaining the shared secret $p^{x_{a}}\;\mathrm{mod}\;n$. Unfortunately, this version of the protocol is also vulnerable to a division attack [5].3.*Non-invertible KEP.* This protocol defines a public key exchange algorithm. To surpass the division attack, ni-KEP introduces Euler’s identity to derive the keys which are defined according to the relations $\left\{p^{x_{i}}k_{i}\;\mathrm{mod}\;n,\;q^{y_{i}}k_{i}\;\mathrm{mod}\;n\right\}$, i=a,b for Alice’s and Bob’s public keys respectively and *n* is obtained as n=p·q·r where *p* and *q* are small prime public numbers and *r* is a big prime public integer. On the other hand, $\left\{k_{i},x_{i}\right\}$ constitute the private key, while the number $y_{i}$ is derived from the equation $\phi \left(n\right)=x_{i}+y_{i}+1$ where ϕ(n) is the Euler’s totient equation. A detailed discussion of this protocol will be presented in a later section.

The public keys of the ni-KEP (and also the cipher texts) exhibit perfect indistinguishability [5]. It means, in the first case, that every $k_{i}$ in the ring satisfies the public key (exists a corresponding $x_{i}$). In the second case, it implies that each ciphertext $c_{i}$ can be derived by any $k_{i}$ in the ring (exists a corresponding $m_{i}$). In view of the above, we claim that the unique opportunity for the eavesdropper, in order to get the private key (or the plaintext), is implementing an exhaustive search among the elements in the ring which is equivalent to searching an unsorted database problem.

Consider symmetric cryptography, which is assumed to be post-quantum because a quantum computer running Grover’s algorithm requires computational cost proportional to the square root of the key size which takes $O(\sqrt{N})$ time. Despite this, an adjustment in the key size prevents the crypto system of being vulnerable to Grover’s algorithm which is the fastest possible quantum algorithm for searching an unsorted database. By contrast, a classical computer requires a linear search, which is O(N) in time to find the same entry [6].

For this reason, we claim that our algorithm is post quantum. On the other hand, we do not devise how Shor’s algorithm would be used to break this protocol. As a consequence, Lizama’s key sizes must be carefully chosen to resist a hypothetical quantum computer running Grover’s algorithm.

**Our contribution.** In this work, we enhance Lizama’s non-invertible key exchange method [5] in order to support Certification Authorities (CA) to allow users to exchange digital certificates which are bounded to their public keys. We claim that our cryptosystem exhibits competitive key size and is able to handle certificated keys, interdomain certification and perfect forward secrecy.

**Organization of the paper.** First, in Section 2, we discuss the main quantum cryptographic approaches: quantum and post-quantum. In Section 3, we summarize principles of public key cryptography considering digital certificates and the Certification Authority role. Then we describe in Section 4 Lizama’s non-invertible protocol to put forward and in Section 5 how Lizama’s KEP can be used to support CAs in single and multiple certification domains. Finally, Section 6 explains a method to derive a new session key from a past session key, thus achieving Perfect Forward Secrecy (PFS). Appendix A contains a brief description about RSA and DH cryptosystems along two possible attacks: prefix and multiplication-based attacks.

## 2. Cryptography in the Quantum Era

Cryptography in the quantum era can be classified into two main approaches: quantum and post-quantum cryptography. A formal discussion of such approaches is beyond the scope of the present article. Let us simply mention that quantum cryptography relies on quantum physics principles that allow to establish a secret key between two authenticated remote parties [7]. The eavesdropper cannot control quantum communication because it produces a detectable noise. Works have been done recently to resist quantum attacks [8,9,10].

On the other side, post-quantum cryptography encompasses cryptographic mathematical methods conceived to resist computational capacity of quantum computers [4,11]. Several methods have been formulated based on computational problems whose complexity surpass the theoretical capacities of quantum computers. Not wishing to fully cover all cases, most promissory techniques include lattices, supersingular isogeny, multivariate equations, code and hash-based cryptography.

Lattice-based methods have demonstrated good performance, by generating short ciphertext, short keys and short signatures [12,13]. Similar to Diffie–Hellman key exchange is the Supersingular Isogeny Diffie–Hellman (SIDH) method which is a quantum resistant key exchange algorithm [14,15]. Supersingular Elliptic Curve Isogeny Cryptography (SIDH) produces very small key sizes but it shows slower performance. The representative algorithm is the Supersingular Isogeny Key Encapsulation (SIKE). The basic objects of multivariate cryptography are systems of nonlinear (usually quadratic) polynomial equations in several variables over a finite field. When performing a digital signature, the set of equations constitute the public key. The receiver computes the hash to verify that the output of the equations corresponds to the hash of the message that is signed [16]. A code-based cryptosystem is essentially a form of error correction code. The private key is a code C, which allows to correct t errors. The sender will encode the message with the public key and include t errors during encoding, then the ciphertext is obtained by adding an error vector to each codeword. With code C, the receiver will be able to accurately correct the errors when decoding the message. Hash-based cryptography was introduced by Lamport, later it was enhanced using Merkle trees [17] and Lizama’s hash-based methods [18,19].

## 3. Public Key Cryptography

### 3.1. Digital Certificates

A cryptographic certificate is basically, a verified public key signed by a third trusted party called Certification Authority (CA). By using this method, each user can verify the origin of a request before accepting it. The importance of a certified key can be illustrated showing a man in the middle (MITM) attack over the Diffie–Hellman (DH) protocol, the first public key exchange algorithm [20]. In Figure 1, we represent the steps required for this key exchange algorithm where the integer prime *p* and *g* are publicly known. A description in depth can be found in Appendix A.

Since there is no method to verify the origin of the integer numbers exchanged across the public channel, an eavesdropper can implement a man in the middle (MITM) attack over the Diffie–Hellman method as it is observed in Figure 2.

To avoid an MITM attack over DH protocol, the RSA algorithm can be added to the exchange protocol. RSA is described in Appendix A. Another common method to protect DH key exchange algorithm is elliptic-curve cryptography [21,22], however Lizama’s protocol is closely related to RSA, thus we describe here RSA and DH.

Figure 3 shows that Alice encrypts the DF constructor $g^{x_{a}}\;\mathrm{mod}\;p$ with Bob’s public key written as $\left(e_{b},n_{b}\right)$, so that only Bob can decrypt it using his private key $d_{b}$. Alice verifies the received message because it is attached a hash of the secret key computed by Bob as represented in Figure 3.

In order for Alice to verify Bob’s public key, provided it does not come from an illegitimate user, Bob must register first his public key with the Certification Authority abbreviated as CA (a third trusted party). Generally speaking, Bob obtains a certificate of his public key $C_{B}$ after CA encrypts (signing) Bob’s public key with CA’s private key $P_{R_{CA}}$. In the next relations, encryption (or decryption) process is denoted as square brackets while the encryption (or decryption) key is outside the brackets:$$C_{B}={\left[P_{U_{B}}\right]}_{P_{R_{CA}}}$$

Every user can obtain and verify Bob’s public key decrypting $C_{B}$ with CA’s public key $P_{R_{CA}}$:$$P_{U_{B}}={\left[C_{B}\right]}_{P_{U_{CA}}}$$

### 3.2. Certification Authority (CA)

As mentioned earlier, a Certification Authority (CA) is a trusted third party that signs a user public key using CA’s private key therefore binding the subject’s identity (and associated information including the name of the owner) to the user’s public key inside a cryptographic certificate. Cryptographic certificates can be exploited to achieve digital signatures in a wide broad of internet transactions and PKI: certificates (X.509), secure channels (TLS) and email (S/MIME).

In view of the imminent arrival of quantum computers, it is unpostponable to develop strategies to adapt the public key infrastructure (PKI) for transition to the quantum era [3,4]. Up to now, few works have been published that adapt existing certificates to quantum certificates or hybrid certificates, which include two public keys for the subject, one classical and one post-quantum and two CA signatures [23,24]. Other works have evaluated existing mechanisms to deal with large records like record fragmentation, segmentation, caching and compression [25]. One the main challenges reported is the difficulty to manage larger certificates by some cryptographic software libraries.

ITU-T Recommendation X.509 defines the format of public key certificates as well as the provision of authentication services under a centralized control scheme that is represented by a directory [26,27]. X.509 assumes a hierarchical system of Certificate Authorities (CAs) for issuing certificates. This contrasts with web of trust models, like PGP, where users sign others’ key certificates to establish the authenticity of the binding between a public key and its owner [28].

A PKI is arranged hierarchically, so that there is always a direct path (a certificate chain) from the root CA to every end-entity. Therefore, with many users, it may be more practical to have a series of CAs, each of which securely provides its public key to a fraction of the users.

If Alice has a certificate from CA${}_{1}$ and Bob owns a certificate from CA${}_{2}$ but Alice does not securely know the public key of CA${}_{2}$, then Bob’s public certificate emitted by CA${}_{2}$, cannot be used by Alice. However, if the two CAs have securely exchanged their own public keys, the following procedure will enable Alice to obtain Bob’s public key:1.Alice obtains the certificate of CA${}_{2}$ signed by CA${}_{1}$. Since Alice has the public key of CA${}_{1}$, she can get the public key of CA${}_{2}$ from its certificate and verify it using the signature of CA${}_{1}$ on the certificate.2.From the directory, Alice obtains the certificate of Bob signed by CA${}_{2}$. Since Alice now has the public key of CA${}_{2}$, she can verify the signature, therefore getting Bob’s public key.

## 4. Lizama’s Key Exchange Protocol

Lizama’s key exchange protocol was introduced in [5], there it can be found all details about the method and its security. The protocol is illustrated in Figure 4. The public key of user *i* (*a* for Alice, *b* for Bob) has two components $\left(P_{i},Q_{i}\right)$ where $P_{i}=p^{2x_{i}}k_{i}\;\mathrm{mod}\;n$ and $Q_{i}=q^{y_{i}}k_{i}\;\mathrm{mod}\;n$. The value $x_{i}$ is chosen randomly while $y_{i}=\phi \left(n\right)-x_{i}+1$. The module *n* is the product of tree public integer primes, so that n=p·q·r where *p* and *q* are small integer primes and *r* is a big integer prime. To achieve indistinguishability *p* and *q* are suggested to be 2, since 2 is a primitive root module *r* (see [5]). The exponent is chosen to be $2x_{i}$ instead of $x_{i}$ to avoid a multiplication attack (see Appendix A). The $x_{i}$ value constitutes along $k_{i}$ the private key of user *i* where $k_{i}$ is an invertible integer in the ring. Users share their public keys $\left(P_{a},Q_{a}\right)$ and $\left(P_{b},Q_{b}\right)$ as well as the integer module *n*. The steps of the protocols are summarized as follows:1.Once public keys have been exchanged, Alice and Bob perform two operations over the numbers received: exponentiation and multiplication as indicated in Table 1.2.To derive the results in the right column of Table 1, Euler’s theorem is applied in $Z_{n}$. The theorem is written in Equation (Equation 1) where *r* is an integer safe prime. As a result that n=pqr, we have that ϕ(n)=(p−1)(q−1)(r−1). Here, *k* and *n* are relative prime to each other, so *k* is an invertible integer in $Z_{n}$. The exponent $x_{i}$ constitutes the private key, is chosen randomly, but $x_{i}$ and $y_{i}$ sum up ϕ(n)+1, thus according to Equation (Equation 1) we have $k^{\phi (n)+1}=k^{\phi (n)}\cdot k^{1}=k$ because *k* is an invertible integer in $Z_{n}$.
(1)$$k^{\phi (n)}\equiv 1\;\mathrm{mod}\;n$$3.Users exchange the resulting value $p^{2x_{a}x_{b}}q^{y_{a}y_{b}}k_{i}\;\mathrm{mod}\;n$, which is multiplied by the corresponding inverse ${k_{i}}^{-1}$ at each side to derive the secret shared key $p^{2x_{a}x_{b}}q^{y_{a}y_{b}}\;\mathrm{mod}\;n$ as depicted in Figure 4.

**Figure 4 entropy-23-00226-f004:**
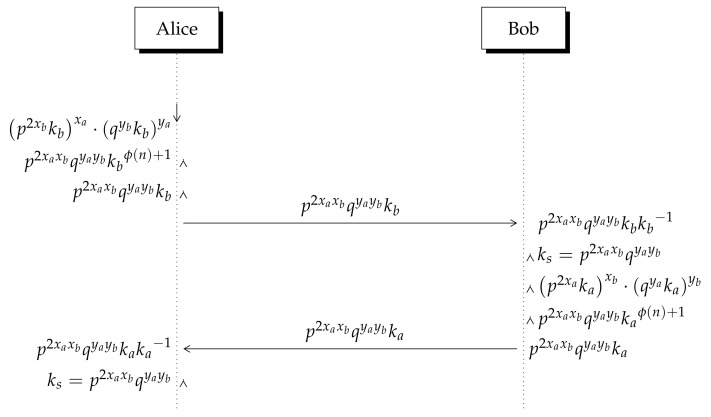
Lizama’s non-invertible key exchange method (KEP) [5]. All operations are modulo *n* where n=pqr. According to Euler’s theorem $k^{\phi (n)+1}\;\mathrm{mod}\;n=k$ because *k* is an invertible integer in $Z_{n}$.

As an example of the required bits for the keys, consider that case where p=q=2 and |r|=1024 (the symbol || denotes the number of bits) the length of the private key yields 1536 bits (|x|=512 and |k|=1024) while the public key (P,Q) contains 2056 bits [5]. In this case, the security level of the secret key is 1024. The process to determine the size of the key is the following: $P=p^{2x}\cdot k\;\mathrm{mod}\;n$ thus $P=p^{2x}\;\mathrm{mod}\;n\cdot k\;\mathrm{mod}\;n$, which in turn implies that:—|k|=|n|—if p=2 and n=4r, we have $2^{2x}\;\mathrm{mod}\;4r$, then $4^{x}\;\mathrm{mod}\;4r$ yields |4|·|x|=|4|+|r| and $\left|x\right|∼\frac{\left|r\right|}{2}$.—since the private key is conformed by *x* and *k*, its size is computed as |n|+|x|∼|r|+|x| which gives 1536.

### 4.1. Cipher-System

In Figure 4, the secret shared key $k_{s}$ is a non-invertible number in $Z_{n}$, thus a convenient method to achieve a cipher-system and secret communication is to divide $k_{s}=p^{2x_{a}x_{b}}q^{y_{a}y_{b}}\;\mathrm{mod}$n by pq, so if we choose p=q=2, then $k_{r}=p^{2x_{a}x_{b}-2}2^{\left(2r-1-x_{a}\right)\left(2r-1-x_{b}\right)}$modr. Now, Alice and Bob can compute its multiplicative inverse ${k_{r}}^{-1}$. Table 2 shows that the enciphered message is obtained as $c=m\cdot k_{r}\;\mathrm{mod}\;r$ and the original plaintext is recovered through the relation $m=c\cdot {k_{r}}^{-1}\;\mathrm{mod}\;r$ because $m=m\cdot k_{r}{k_{r}}^{-1}\;\mathrm{mod}\;r$. To send a message encoded as an integer in $Z_{r}$, the number *m* must be less than *r*.

### 4.2. Mathematical Representation

In the rest of the paper we will use the following mathematical notation: $\left(P_{i},Q_{i}\right)$ which constitutes the public key of user *i*. As stated before, $P_{i}=p^{2x_{i}}k_{i}$ and $Q_{i}=q^{y_{i}}k_{i}$ where $\left(x_{i},k_{i}\right)$ constitutes the private key of user *i* and $x_{i}+y_{i}=\phi \left(n\right)+1$. As stated before, user *j* raises the public key of *i* to its private key. Then, *j* returns to *i* the integer number $\left[k_{i,j}\right]\;k_{i}$ where $\left[k_{i,j}\right]=p^{2x_{i}x_{j}}q^{y_{i}y_{j}}$ and $k_{i}$ is a component of the private key of user *i*, then they apply the inverse of $k_{i}$ in order to derive the shared secret key $k_{i,j}$. The same procedure is applied in the opposite direction so user *i* sends to *j* the integer $\left[k_{i,j}\right]\;k_{j}$ to get the secret number $k_{i,j}$ (see Table 3).

## 5. Key Certification with Lizama’s ni-KEP

In this section, we explain the public key certification method so that a Certification Authority (CA) can certify the user’s public keys using Lizama’s ni-KEP. The protocol steps are as follows:1.To certify their public key with the Certification Authority CA, user *i* sends to CA their public key $\left(P_{i},Q_{i}\right)$.2.If CA approves the request of *i*, they generate and publish the certified key $\left[k_{i,ac}\right]$$k_{i}$ which has been derived according to Table 3.3.The CA’s public database of certified keys can be seen in Table 4 which contains the certified keys of Alice and Bob.

Now, Alice and Bob can establish a secret key with certified keys, but first Alice must download Bob’s certified key from CA’s database and vice versa. The steps to derive the key are depicted in Figure 5 and described as follows:1.Using CA’s public key $\left(P_{ca},Q_{ca}\right)$, Alice computes $\left[k_{a,ca}\right]$$k_{ca}$. In addition, she computes $\left[k_{a,b}\right]$$k_{b}$ using Bob’s public key $\left(P_{b},Q_{b}\right)$.3.Alice multiplies them by Bob’s certified key $\left[k_{b,ca}\right]k_{b}$ and sends the resulting integer number to Bob. The same procedure is applied by him.4.Bob multiplies the received integer by ${k_{b}}^{-1}$ twice, thus he obtains the secret shared key $K_{ab}=\left[k_{a,b}\right]\left[k_{b,ca}\right]\left[k_{a,ca}\right]k_{ca}$ (see Figure 5).5.Applying this procedure, Bob derives the same secret number $K_{ab}$.

It must be highlighted that in order to establish the secret key the certified key of the intended user must be applied but also the public key of the Certification Authority CA. Moreover, each user must apply (twice) their private key to get the shared secret key. In addition, to avoid a prefix attack the relation $K_{ab}>r$ must be satisfied (see Appendix A).

**Figure 5 entropy-23-00226-f005:**
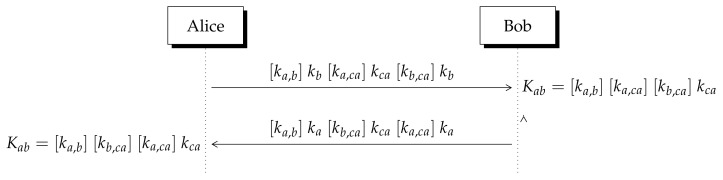
Non-invertible KEP with Certification Authority (CA). All operations are performed module *n*.

### 5.1. Indistinguishability

An important security property of the ni-KEP is the indistinguishability of $k_{i}$ in the public key integers. It implies that each invertible $k_{i}$ in $Z_{n}$ satisfies the public key along the appropriate $x_{i}$ value. The same property can be deduced for the cipher text, thus every $k_{i}$ in the ring can be used to produce a given cipher text with a specific $m_{i}$.

Indistinguishability can be extended to the certified key exchange method. Let us rewrite the exchanged messages depicted in Figure 5 as $M_{b}\cdot k_{b}\;\mathrm{mod}\;m$ from Alice to Bob so that $M_{b}=\left[k_{a,b}\right]k_{b}\left[k_{a,ca}\right]k_{ca}\left[k_{b,ca}\right]$. Similarly, in the reverse direction we have $M_{a}\cdot k_{a}\;\mathrm{mod}\;n$ which implies that $M_{a}=\left[k_{a,b}\right]k_{a}\left[k_{b,ca}\right]k_{ca}\left[k_{a,ca}\right]$. Applying division to $M_{a}$ (or $M_{b}$) by pq we obtain:$${\left(pq\right)}^{-1}M_{i}\cdot k_{i}\;\mathrm{mod}\;r$$

From here, we know that $M_{i}\;\mathrm{mod}\;r$ and $k_{i}\;\mathrm{mod}\;r$ are integers in $Z_{r}$. Moreover, the multiplication $M_{i}\cdot k_{i}\;\mathrm{mod}\;r$ produces a permutation of the integers in $Z_{r}$ because *r* is an integer prime, thus the resulting integer is in $Z_{r}$. As it was shown in [5], $k_{i}$ remains indistinguishable inside encrypted messages; therefore, the unique opportunity for the eavesdropper is to find the secret key $k_{i}$ by exhaustive search.

### 5.2. Multiple CAs

Suppose Alice has been registered with CA${}_{1}$ while Bob has a certified key from CA${}_{2}$. In addition, Alice receives from Bob its certified key and vice versa but Alice does not have access to CA${}_{2}$’s database neither Bob to CA${}_{1}$’s database. As indicated in Table 5, CA${}_{1}$’s database is accessible to Alice and CA${}_{2}$’s database is reachable by Bob. However, as can be seen there, CA${}_{1}$’s database contains the certified key of CA${}_{2}$ and CA${}_{2}$’s database contains the certificate of CA${}_{1}$. Then, they follow the steps depicted in Figure 6 and detailed below:1.Using CA${}_{1}$’s public key $\left(P_{ca_{1}},Q_{ca_{1}}\right)$, Alice computes $\left[k_{a,ac_{1}}\right]$$k_{ac_{1}}$, she also computes $\left[k_{a,b}\right]$$k_{b}$ with Bob’s public key $\left(P_{b},Q_{b}\right)$.3.Alice multiplies them by Bob’s certificate $\left[k_{b,ca_{2}}\right]k_{b}$ and CA${}_{2}$’s certificate $\left[k_{ca_{1},ca_{2}}\right]$$k_{{ca}_{2}}$ and sends the resulting integer number to Bob. The same procedure is applied by Bob.4.Alice multiplies the received integer by ${k_{a}}^{-1}$ twice, thus she obtains the secret shared key $K_{ab}=\left[k_{a,b}\right]\left[k_{a,{ca}_{1}}\right]k_{{ca}_{1}}\left[k_{b,{ca}_{2}}\right]k_{{ca}_{2}}\left[k_{ca_{1},ca_{2}}\right]$ (see Figure 6).5.Applying the same procedure, Bob derives the secret shared number $K_{ab}$.

**Table 5 entropy-23-00226-t005:** Public databases of CA${}_{1}$ and CA${}_{2}$ which would be located distantly, so database of CA${}_{1}$ is accessible to Alice and CA${}_{2}$’s database is close to Bob.

CA	User	Public Key	Certified Key
	CA${}_{1}$	$\left(P_{ca_{1}},Q_{ca_{1}}\right)$	-
	CA${}_{2}$	$\left(P_{ca_{2}},Q_{ca_{2}}\right)$	$\left[k_{ca_{1},ca_{2}}\right]$ $k_{{ca}_{2}}$
CA${}_{1}$	Alice	$\left(P_{a},Q_{a}\right)$	$\left[k_{a,{ca}_{1}}\right]$ $k_{a}$
	CA${}_{2}$	$\left(P_{ca_{2}},Q_{ca_{2}}\right)$	-
	CA${}_{1}$	$\left(P_{ca_{1}},Q_{ca_{1}}\right)$	$\left[k_{ca_{1},ca_{2}}\right]$ $k_{{ca}_{1}}$
CA${}_{2}$	Bob	$\left(P_{b},Q_{b}\right)$	$\left[k_{b,{ca}_{2}}\right]$ $k_{b}$

**Figure 6 entropy-23-00226-f006:**
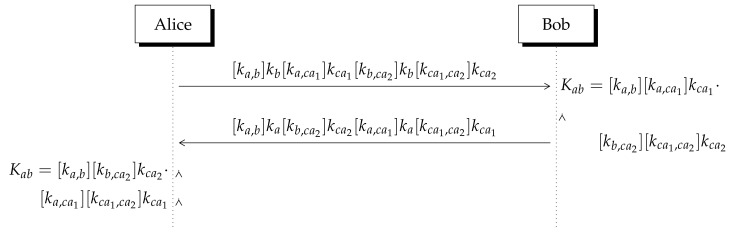
Non-invertible KEP with two CAs. Operations are performed module *n*.

## 6. Perfect Forward Secrecy (PFS)

Suppose Alice and Bob require to establish a new confidential communication. However, they do not want to use the same secret key of the last session. Perfect forward secrecy (PFS) is a feature of key agreement protocols that guarantee that, if the currently key was compromised, it does not compromise the security of previously used keys. Therefore, the security of encrypted messages using old keys persists. When a system has a perfect forward secret, the system is said to be forward secure.

In the next procedure, we demonstrate that Lizama’s non-invertible KEP is enhanced to exhibit PFS (see Table 6 and Figure 7).

1.Alice and Bob share a certified key $K_{i}$ from a previous exchange.2.Using CA’s public key $\left(P_{ca},Q_{ca}\right)$, Alice computes $\left[k_{a,ca}\right]$$k_{ca}$. In addition, according to Table 6, Alice computes ${\left[k_{a,b}\right]}^{K_{i}}$${k_{b}}^{K_{i}}$ using Bob’s public key $\left(P_{b},Q_{b}\right)$.4.Alice multiplies them by Bob’s certificate $\left[k_{b,ca}\right]$$k_{b}$ and sends the resulting number to Bob. The same procedure is applied by Bob.5.Bob multiplies the received integer by ${k_{b}}^{-K_{i}-1}$, thus he obtains the secret shared key $K_{i+1}={\left[k_{a,b}\right]}^{K_{i}}\left[k_{a,ca}\right]\left[k_{b,ca}\right]k_{ca}$ (see Figure 7).6.Conversely, Alice multiplies the received integer by ${k_{a}}^{-K_{i}-1}$, thus she gets the secret shared key $K_{i+1}={\left[k_{a,b}\right]}^{K_{i}}\left[k_{b,ca}\right]\left[k_{a,ca}\right]k_{ca}$.

Therefore, the eavesdropper cannot derive $K_{i}$ from $K_{i+1}$ and the procedure can be repeated as many times as required to derive $K_{m+1}$ from $K_{m}$.

**Table 6 entropy-23-00226-t006:** Mathematical operations to achieve perfect secrecy (PFS).

Short Notation	Mathematical Operation
$\left(P_{i},Q_{i}\right)$	$\;P_{i}=p^{2x_{i}}k_{i}$, $Q_{i}=q^{y_{i}}k_{i}$
${P_{i}}^{x_{j}}\cdot {Q_{i}}^{y_{j}}$	$\;{\left(p^{2x_{i}}k_{i}\right)}^{x_{j}}\cdot {\left(q^{y_{i}}k_{i}\right)}^{y_{j}}$
$\left[k_{i,j}\right]$ $k_{i}$	$\;p^{2x_{i}x_{j}}q^{y_{i}y_{j}}k_{i}$
${P_{i}}^{k_{s}x_{j}}\cdot {Q_{i}}^{k_{s}y_{j}}$	$\;{\left(p^{2x_{i}}k_{i}\right)}^{k_{s}x_{j}}\cdot {\left(q^{y_{i}}k_{i}\right)}^{k_{s}y_{j}}$
${\left[k_{i,j}\right]}^{k_{s}}$ ${k_{i}}^{k_{s}}$	$\;p^{2k_{s}x_{i}x_{j}}q^{k_{s}y_{i}y_{j}}{k_{i}}^{k_{s}}$

**Figure 7 entropy-23-00226-f007:**
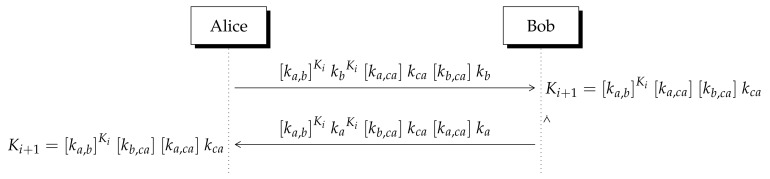
Alice and Bob require to establish a new secret key $K_{i+1}$. However, they do not want to use the last secret key $K_{i}$. This procedure is repeated to derive $K_{i+2}$ from $K_{i+1}$.

## 7. Discussion

After the third evaluation round, NIST has selected seven algorithms (and eight alternative candidates), four of them are public key encryption (and key-establishment) systems and three correspond to digital signature algorithms. In the first category, CRYSTALS-KYBER, NTRU-HPS, SABER are lattice-based while Classic McEliece is a code-based public key encryption system. Regarding digital signature schemes, CRYSTALS-DILITHIUM and FALCON are lattice-based and Rainbow is a multivariate-based algorithm. Since we are only concerned with the first category, we found that public keys size in Lizama’s protocol has the smallest size: 0.256 kilobytes (for |n|=1024) while the corresponding certified key size achieves 0.384 kilobytes (see Table 7). Furthermore, to reduce the required storage space a good strategy would be saving only one component of the public key $\left(P_{i},Q_{i}\right)$, e.g., $P_{i}$ while the second one $Q_{i}$ will be transferred directly from Alice to Bob. In that case the certified public key size decreases from 0.384 to 0.256 KB.

We emphasize the importance of the key size because, as it was shown in [25], the key size of known quantum-resistant schemes can grow from a few to many kilobytes which can arise some difficulties for today’s existing infrastructures of X.509 certificates. A good example is that post-quantum TLS handshake takes 40 KB which is 24 times more expensive [30]. Even worse, there would be some scenarios which are very sensitive to delays that cannot store big certificates or perform signature (generation or verification) because of those limitations. In such scenarios, most post-quantum signatures would be impractical because of their required computational cost.

Still under study is the Identity-Based Encryption (IBE) scheme which is considered an alternative to traditional certificate-based public key cryptography to reduce communication overheads in wireless sensor networks. In [30], it has been found that ID-based TLS is 2.8× costlier than certificate-based TLS in the pre-quantum scenario.

## 8. Conclusions

We have detailed the steps to enhance the Lizama’s non-invertible key exchange protocol to be used as a public key cryptosystem with single and multiple certification domains. We have provided the specification the certification authority keys and the method to certify the user’s public keys. Therefore, our approach is scalable and interoperable and can be exploited in the pre-quantum and the quantum era because the protocol exhibits indistinguishability of the integers in the public key and ciphertexts.

We found that public keys size in Lizama’s protocol has the smallest size regarding main post-quantum systems: 0.256 kilobytes and 0.384 kilobytes for public key and certified key, respectively. Moreover, we suggest that the public key database only stores one component of the two integers which are part of the public key, while the second component can be transferred directly to the remote destination. This strategy reduces the required storage space of certified keys to 0.256 kilobytes. Therefore it makes manageable some issues caused by large certificates as fragmentation, segmentation and caching.

Furthermore, we have discussed a method to achieve perfect forward secrecy (PFS) so that a session key can be derived from the previous one and the procedure is repeated as many times as necessary. 

## Figures and Tables

**Figure 1 entropy-23-00226-f001:**
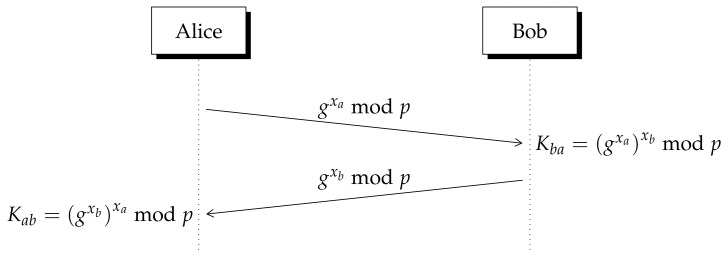
Basic Diffie–Hellman protocol. All operations are performed module *p*.

**Figure 2 entropy-23-00226-f002:**
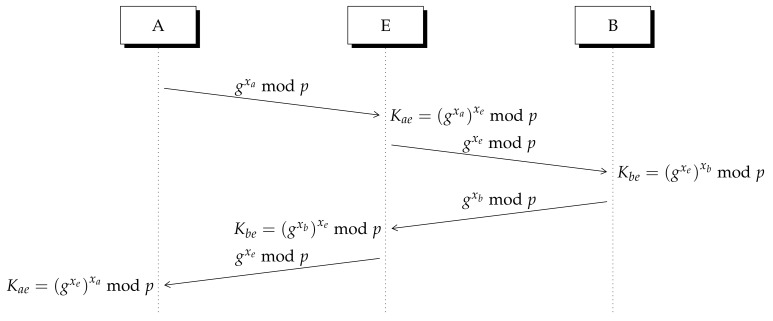
A man in the middle (MITM) attack over Diffie–Hellman (DH) protocol. The eavesdropper obtains a key with Alice $K_{ae}$ and other with Bob $K_{be}$. Legitimate users cannot verify the origin of exchanged numbers.

**Figure 3 entropy-23-00226-f003:**
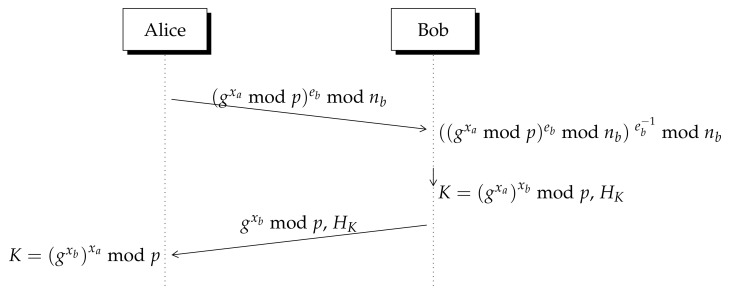
Diffie–Hellman algorithm with RSA. Bob’s public key is written as $P_{U_{B}}=\left(e_{b},n_{b}\right)$, Bob’s private key is ${e_{b}}^{-1}$ that indicates the inverse of $e_{b}$ in $Z_{\phi (n)}$. $H_{K}$ represents the hash value of *K* which is used by Alice to verify the origin of the received number.

**Table 1 entropy-23-00226-t001:** These operations (exponentiation and multiplication) are performed at each side after public keys of users are exchanged.

User	Operation	Result
Alice	$\;\;{\left(p^{2x_{b}}\cdot k_{b}\;\mathrm{mod}\;n\right)}^{x_{a}}\cdot {\left(q^{y_{b}}\cdot k_{b}\;\mathrm{mod}\;n\right)}^{y_{a}}$	$p^{2x_{b}x_{a}}q^{y_{b}y_{a}}\cdot k_{b}\;\mathrm{mod}\;n$
Bob	$\;\;{\left(p^{2x_{a}}\cdot k_{a}\;\mathrm{mod}\;n\right)}^{x_{b}}\cdot {\left(q^{y_{a}}\cdot k_{a}\;\mathrm{mod}\;n\right)}^{y_{b}}$	$p^{2x_{a}x_{b}}q^{y_{a}y_{b}}\cdot k_{a}\;\mathrm{mod}\;n$

**Table 2 entropy-23-00226-t002:** Lizama’s key exchange algorithm can be used to encrypt/decrypt messages provided $k_{s}$ is divided by pq.

Mode	Mathematical Relation
Encryption	$c=m\cdot k_{r}\;\;\mathrm{mod}\;\;r$
Decryption	$m=c\cdot {k_{r}}^{-1}\;\;\mathrm{mod}\;\;r$

**Table 3 entropy-23-00226-t003:** Mathematical representation. All operations are performed module *n*.

Short Notation	Mathematical Operation
$\left(P_{i},Q_{i}\right)$	$P_{i}=p^{2x_{i}}k_{i}$, $Q_{i}=q^{y_{i}}k_{i}$
${P_{i}}^{x_{j}}\cdot {Q_{i}}^{y_{j}}$	${\left(p^{2x_{i}}k_{i}\right)}^{x_{j}}\cdot {\left(q^{y_{i}}k_{i}\right)}^{y_{j}}$
$\left[k_{i,j}\right]$ $k_{i}$	$p^{2x_{i}x_{j}}q^{y_{i}y_{j}}k_{i}$

**Table 4 entropy-23-00226-t004:** CA’s public database. The Certification Authority CA publishes their public key $\left(P_{ca},Q_{ca}\right)$.

User	Public Key	Certified Key
CA	$\left(P_{ca},Q_{ca}\right)$	-
Alice	$\left(P_{a},Q_{a}\right)$	$\left[k_{a,ca}\right]$ $k_{a}$
Bob	$\left(P_{b},Q_{b}\right)$	$\left[k_{b,ca}\right]$ $k_{b}$

**Table 7 entropy-23-00226-t007:** A comparison of Lizama’s protocol against National Institute of Standards and Technology (NIST) Round 3 finalists is shown in the categories of public key encryption and key-establishment algorithms [29].

Scheme	System	Public Key (KB)	Private Key (KB)	Signature (KB)
Public Key/ KEM	LIZAMA’S KEP	0.256–0.512	0.192–0.384	–
	Classic McEliece	261,120–1,357,824	6492–14,120	–
	CRYSTALS-KYBER	1.632–3.168	0.8–1.568	–
	NTRU-HPS	0.931–1.230	1.235–1.592	–
	SABER	0.672–1.312	1.568–3.040	–
Signature Algorithms	CRYSTALS-DILITHIUM	1.312–2.592	–	2.420–4.595
	FALCON	0.897–1.793	–	0.666–0.280
	Rainbow	157.8–1885.4	101.2–1375.7	0.066–0.212

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
