# Peer review of "Non-Invertible Public Key Certificates"

_entropy, 2021, doi:10.3390/e23020226_

Round 1

Reviewer 1 Report

The authors describe a public certification system that is based on Lizama's non-invertible key exchange protocol to implement a PKI for post-quantum cryptosystems. The authors should provide a paragraph describing their contribution and the novelty of the proposed solution. At the same time, it would be good to provide a table mentioning the differences of the proposed solution to the ones already existing.

Author Response

The authors deeply value the comprehensive comments and suggestions to perform this revision. Please find below our reply to the points raised and the corresponding modifications we have made in the revised manuscript to accommodate all the reviewers’ comments.

Reviewer 2 Report

This paper proposed an enhancement of Lizama’s non-invertible key exchange method to be completely functional for PKC including Certification Authorities (CA). Furthermore, this paper also proposes a new method to achieve Perfect Forward Secrecy (PFS).

First of all, I think that the introduction can be improved much better. For example, from paragraph 2 to 3, there is a big gap that requires some filled in. More explicitly, the authors can talk about what is Lizama’s non-invertible key exchange method, what is its importance and applications. The same applies for CA and PFS. Furthermore, there is no clear indications/summary of the content of the paper in the introduction, i.e. what does each section do.

I will say that the organization of the paper is badly arranged. For instances, Section 4 and Section 6 have the same section title “Certification Authority”, which do not describe the content in the section precisely.

The arguments and constructions to achieve PFS proposed in this paper seems correct. However, I strongly suggest the authors to do thorough editing of their paper in terms of presentation style and English grammar.

Finally, one thing that puzzles me is: how does post-quantum cryptography come into play in this paper? From the arguments in the paper, I do not see the relationship and relevance of post-quantum cryptography to the PKI and the proposed method to achieve PFS. Hopefully the authors will address this issue in their revised paper.

Author Response

(The authors gave the same response as above.)

Reviewer 3 Report

This paper presents a public certification system based in Lizama’s non-invertible Key Exchange Protocol which can be used to implement a public key infrastructure (PKI). The authors also show functionality of certificates across different certification domains, and discuss that non-invertible certificates can exhibit PFS. Some descriptions need to be improved and revised.

  1. Compared with related works, the advantages of the proposed scheme could be described in more detailed.
  2. The reason why the proposed scheme is based on Lizama’s non-invertible Key Exchange Protocol should be clearly described.
  3. In addition to PFS, more analyses about the proposed scheme could be included, such as correctness, authentication and unforgeability.

Author Response

(The authors gave the same response as above.)

Round 2

Reviewer 1 Report

The authors have addressed the reviewer comments satisfactory.

I invite them to check Section 7 - Discussions since in some references instead of the number a question mark appears in the text. Also, the Conclusions section is very weak.

Author Response

We truly thank Reviewer # 1 for his comments, remarks and suggestions. We have made the best effort to take into account each of the comments and we hope that our corrections/adjustments are adequate to attend to suggestions sufficiently.

Reviewer 2 Report

Overview and general recommendation:

This paper is a revision of the paper which proposed an enhancement of Lizama’s non-invertible key exchange method to be completely functional for PKC including Certification Authorities (CA). Furthermore, this paper also proposes a new method to achieve Perfect Forward Secrecy (PFS).

I think most of the comments given by the reviewers are addressed accordingly, except on the issue of comparison and parameter. I think the authors should try to describe their method to calculate the key size of their cryptosystem, as they claimed that “our cryptosystem reduces the size of the keys and is able to handle certificated keys, interdomain certification and perfect forward secrecy”.

Also, I think the key size comparisons in Table 7 is incomplete. More specifically, when they compare the size with code-based, lattice-based, and etc, which schemes do they compare to? In each of the code-based, lattice-based (and etc) approach, there are so many schemes out there. Perhaps the authors can consider some of the candidates submitted to the NIST PQC Standardization.

Some other minor comments:

  1. In Abstract: based in Lizama’s -> based on Lizama’s
  2. In Abstract: implement a public key infrastructure … efficient -> implement a secure, scalable, interoperable and efficient public key infrastructure (PKI).
  3. Pg 2, Line 69: For quantum computer, if the search requires N operations, then for classical computer the search requires 2N instead of N/2, i.e. speed of exhaustive search in quantum computer is halved by using Grover’s algorithm
  4. Pg 2, Line 75-87: Please break this paragraph into two paragraphs: first paragraph with short headline “Our Contribution.” and the second paragraph with short headline “Organization of the paper.”
  5. Pg 3, Line 92: in quantum physics -> on quantum physics
  6. Pg 3, Line 94: Work has been done -> Works have been done
  7. Pg 3, Line 103: ciphertext and keys and short signatures -> ciphertext, short keys and short signatures
  8. Pg 3, Line 102-118: I suggest that these two paragraphs can be merged into one, as the content can be viewed as a same category.
  9. Pg 3, Line 131: it can be added the RSA algorithm -> the RSA algorithm can be added
  10. Pg 5, Line 160: traditional -> classical
  11. Pg 10, Line 309: 24 more expensive -> 24 times more expensive. Also, the citation is incomplete with [?]
  12. Pg 10, Line 310: some scenarios very sensitive -> some scenarios which are very sensitive
  13. Pg 10, Line 315: Public keys in -> Public keys size in
  14. Pg 10, Line 316: certified key -> certified key size
  15. Pg 11, Line 323: In [?] – incomplete citation

Author Response

We really thank to the Reviewer #2 and hope that his suggestions have been sufficiently addressed. Please refer to the attached document to find our reply to the points raised and the corresponding modifications we have made to improve the manuscript.

Reviewer 3 Report

The authors have revised their manuscript and solved most previous problems in this version. Some descriptions still need to be revised slightly.

  1. The description of some data on Sec. 7 is inconsistent with Table 7. The authors should make sure them.
  2. On Sec. 7, some reference citations are missed.

Author Response

We really thank Reviewer #3 for his comments and suggestions that have been useful to improve the manuscript and we hope the article will be published in the Special Issue of Entropy.
